# Non-fatal injuries in rural Burkina Faso amongst older adults, disease burden and health system responsiveness: a cross-sectional household survey

John Whitaker [1,2,3] Guy Harling [4,5,6,7] Ali Sie,[8] Mamadou Bountogo,[8] Lisa R Hirschhorn [9] Jennifer Manne-Goehler,[10] Till Bärnighausen,[6,11] Justine Davies[1,7,12]

► Prepublication history and additional online supplemental material for this paper are available online. To view these files, please visit the journal online (http://dx.doi.org/10.1136/bmjopen-2020-045621).

For numbered affiliations see end of article.

**Correspondence to**
John Whitaker;
john.k.whitaker@kcl.ac.uk

## ABSTRACT

**Objectives** This study aimed to evaluate the epidemiology of injury as well as patient-reported health system responsiveness following injury and how this compares with non-injured patient experience, in older individuals in rural Burkina Faso.

**Design** Cross-sectional household survey. Secondary analysis of the CRSN Heidelberg Ageing Study dataset.

**Setting** Rural Burkina Faso.

**Participants** 3028 adults, over 40, from multiple ethnic groups, were randomly sampled from the 2015 Nouna Health and Demographic Surveillance Site census.

**Primary and secondary outcome measures** Primary outcome was incidence of injury. Secondary outcomes were incidence of injury related disability and patient-reported health system responsiveness following injury.

**Results** 7.7% (232/3028) of the population reported injury in the preceding 12 months. In multivariable analyses, younger age, male sex, highest wealth quintile, an abnormal Generalised Anxiety Disorder score and lower Quality of Life score were all associated with injury. The most common mechanism of injury was being struck or hit by an object, 32.8%. In multivariable analysis, only education was significantly negatively associated with odds of disability (OR 0.407, 95% CI 0.17 to 0.997). Across all survey participants, 3.9% (119/3028) reported their most recent care seeking episode was following injury, rather than for another condition. Positive experience and satisfaction with care were reported following injury, with shorter median wait times (10 vs 20 min, p=0.002) and longer consultation times (20 vs 15 min, p=0.002) than care for another reason. Injured patients were also asked to return to health facilities more often than those seeking care for another reason, 81.4% (95% CI 73.1% to 87.9%) vs 54.8% (95% CI 49.9% to 53.6%).

**Conclusions** Injury is an important disease burden in this older adult rural low-income and middle-income country population. Further research could inform preventative strategies, including safer rural farming methods, explore the association between adverse mental health and injury, and strengthen health system readiness to provide quality care.

## INTRODUCTION

Injury is a neglected but important cause of avoidable disability and causes more than five

### Strengths and limitations of this study

► Through a random sampling strategy, our household survey was able to establish the incidence of non-fatal injury in rural Burkina Faso, where little empirical data on injury exists.

► By including variables of psychological morbidity and quality of life, we were able to explore associations with those reporting injuries, an understudied aspect of injury burden on low-income settings.

► By establishing the most recent reason for accessing care, we were able to compare the health system responsiveness following injury with other conditions.

► The study was cross-sectional, which limits the causal interpretation of our findings.

► The survey lacked clear definitions of injury and disability, which may have led to an overestimation of burden.

million deaths globally every year,[1] more than tuberculosis, malaria and HIV combined. Low-income and middle-income countries (LMICs) disproportionately bear this burden with 90% of global injury related deaths.[2] However, injury-related deaths are only the tip of the iceberg with an estimated one billion people sustaining injuries that require healthcare annually.[1]

Burkina Faso is a landlocked country of 19 million people in sub-Saharan Africa. It is a low-income country ranked 183 of 189 countries on the Human Development Index[3] with limited natural resources.[3] A 2017 GBD estimates across all age groups that injuries are responsible for 7.32% of deaths and 6.48% of disability-adjusted life-years in Burkina Faso, similar to sub-Saharan Africa rates.[4]

In common with the least developed countries, research investigating injuries in Burkina Faso has been sparse.[5] Studies that have been conducted tend to be referral

facility based often with small case numbers and limited to a single mechanism of injury pattern.[6–10] Road traffic collisions (RTCs) have been investigated specifically, and the WHO has estimated that in 2016 there were 30.5 per 100 000 road traffic fatalities in Burkina Faso, above the average for Africa (26.6) which is the continent with the highest death rate globally.[11] RTC victim data from the tertiary referral hospital in the capital Ouagadougou identified that 87% of road traffic victims attending for tertiary care emergency department were from two wheel motor vehicles; a quarter of these experienced disability beyond 30 days.[12]

Broader injury epidemiology studies from Burkina Faso have primarily studied cause of death data obtained through Verbal Autopsy. In the capital, Ouagadougou, a survey within an urban Health and Demographic Surveillance System (HDSS) identified 4.1% of deaths were due to injury.[13] From the rural Nouna HDSSs Verbal Autopsy data, age-standardised and sex-standardised mortality for external causes of death (the category containing injuries) was almost twice that of the urban comparator in Ougadougou, with the main cause being transport related. Unfortunately, these Verbal Autopsy based surveys do not capture all mechanisms of injury, and they do not allow assessment of non-fatal injury occurrence.[14] Injury has also been characterised as a disease affecting the young, and some population studies of injury in sub-Saharan Africa have even excluded adults over 70.[15] However, older people represent an important and growing population in LMICs. How and why older people are injured, and the consequences associated with these injuries require further exploration.

Injuries can have a lasting impact on the victims through physical disability, previously shown beyond 30 days in over a quarter of RTC victims in Ouagadougou,[12] but also psychological morbidity. From high-income country (HIC) settings, depression, anxiety and post-traumatic stress are commonly associated with physical injury.[16 17] This includes older populations with worse quality of life, psychological and social health status seen following hip fractures and osteoporotic vertebral fractures.[18 19] Poor mental health is also a risk factor for non-accidental injuries.[20 21] The impact of mental health following injury within Burkina Faso among general older adult population has not been studied, with mental health studies limited to vulnerable populations such as sex workers and children exposed to physical violence in Burkina Faso.[22 23]

It has been estimated that if LMIC injury care quality could match that of HIC, then one-third of all trauma deaths could be avoided,[24] It is thus necessary to improve injury epidemiology data from Burkina Faso to inform preventative measures and treatment services. However, the provision of care alone may not be associated with improved outcomes. Such care needs to be responsive to patients needs beyond providing good clinical outcomes in order to engender trust leading to compliance with treatment and encouragement of future injured persons

to attend services.[25–27] However, very few studies on the responsiveness of injury care have been done in LMICs.[28]

This analysis primarily aimed to assess the incidence of non-fatal injury and variables associated with this among older people in rural Burkina Faso, for which little is currently known. Secondary aims were first to describe the incidence of and variables associated with injury related disability, and second, describe patient reported health system responsiveness following injury.

## METHODS
We used the Strengthening the Reporting of Observational Studies in Epidemiology cross-sectional reporting guidelines.[29]

### Study setting
The study was set in the Nouna HDSS area, in the Boucle du Mouhoun region, north-western Burkina Faso. The HDSS collects annual birth, death and migration data in a well-enumerated population. The HDSS area consists of the market town Nouna and 59 surrounding villages with a total population of around 107000.[30] Residents come from multiple ethnic groups, and the major economic activities are farming and animal husbandry. Life expectancy from birth is 58.0 years for men and 61.5 years for women.[31] There is one tarred road running through the area. There are no formal ambulances with emergency transport usually informal via private or taxi motorbike. In rainy season travel can be very difficult.

### Study design
This study is an analysis of the CRSN Heidelberg Ageing Study dataset (CHAS). The study methodology has been described in detail elsewhere.[32 33] Briefly, this cross-sectional study consists of a population-representative sample of adults ≥40 years of age. Three thousand older adults were randomly sampled from the 2015 Nouna HDSS census. In all villages (n=6) with fewer than 50 adults aged over 40, all adults were selected to take part. In all other villages, a random sample of households with at least one person over 40 years old was drawn. Then within each selected household, one age eligible adult was randomly selected to complete the survey, which was administered to them by trained data collectors. Data collection was performed using Open Data Kit software on tablet computers at the participants' residence between May and July 2018.[34] Interviews were conducted either in French or translated into Dioula by the interviewers.

### Variables
The household survey contained questions on age, sex, education, marital status, household assets, experience of injury in past 12 months including mechanism and associated disability, reasons for last health facility visit, and questions covering the WHO health system responsiveness domains[25] derived from other surveys used in sub-Saharan Africa.[35–37] Injury data were self-reported

and injuries were not independently verified. Anxiety was assessed using the Generalised Anxiety Disorder question (GAD-2) score,[38] and depression using Patient Health Questionnaire (PHQ-9).[39 40] Quality of life was measured using the validated EuroHIS 8-item version of WHO Quality of Life (WHOQOL).[41 42] Disability was measured using the 12 item WHO Disability Assessment Schedule, version 2 (WHODAS-II) disability score.[42 43] The Fried frailty score was constructed from questions on weight loss in the past year, self-reported activity and levels of exhaustion, combined with measures of walking speed and grip strength.[44 45]

## Outcome variables

The main outcome variables were whether injured or not in the preceding 12 months, or if injured, whether disabled as a result of the injury. Participants reported whether they had any event where they suffered from bodily injury in the last 12 months. For those reporting yes, the cause of injury was reported along with the question 'did you suffer a physical disability as a result of being injured?'.

## Mechanism of injury

Mechanism of injury was captured as either fall, struck/hit by an object, cut/stabbed, gunshot, fire/heat burn, drowning/near-drowning, poisoning, animal bite, electric shock or other specified by free text. Injury mechanisms with fewer than eight or cases were combined as 'other' for analyses. Those who fell reported whether this was at or higher than ground level.

## Demographic characteristics

Marital status was categorised as married/cohabiting or single/widowed/divorced. Educational level was categorised as no education or any education. Participants were asked 37 questions on household assets and dwelling characteristics; from these, wealth quintiles were derived from the Filmer and Pritchett first principal component method.[46]

## Definitions of disease states

Participants were defined as having symptoms of anxiety based on a GAD-2 score ≥3.[38] Participants scoring 10 or more on PHQ-9 were categorised as having depressive symptoms in this analysis.[40] The calculation of the Fried score used in this study has been described previously.[32] For this analysis participants were dichotomised as robust or prefrail/frail. WHODAS-II disability score was normalised to a 0–100 scale, where 0 equates to no disability and 100 the worst disability. Quality of life[41 42] was similarly normalised to a 0–100 scale, with 100 denoting the best quality of life.

## Health system experience and responsiveness

Regardless of when it occurred, the reason for the most recent episode of health seeking was recorded and classified as either injury or another reason. Participants answering this question were not necessarily the same as those injured in the previous 12 months who may have sought care for another reason subsequent to their injury. There were, therefore, two injury question groups in this study. The first to determine annual injury incidence and characteristics, the second to determine those for whom the last healthcare visit followed an injury. Online supplemental appendix figure 1 illustrates how these overlapping but distinct question groups are reported. Those who had sought care were asked health system responsiveness questions, including: (1) confidence in receiving effective treatment if very sick tomorrow, dichotomised as very/somewhat versus not very/not at all; (2) the overall view of the health system, dichotomised as needs to be rebuilt/major changes needed versus only minor changes needed; (3) trust in the skills and abilities of the healthcare worker at the facility dichotomised as (A). very much, quite a bit or some and (B) very little or not at all; (4) ease or difficulty in following provider's advice dichotomised as 1, very easy, easy or fair and 2. hard or very hard; and (5) opinion of care provider's knowledge and skills, experience of being involved in making decisions for treatment, ability of provider to explain things in a way they could understand and how well the received care met health needs were all dichotomised into positive responses (1) excellent, very good or good and negative responses (2) fair or poor. These variables were dichotomised for ease of interpretation.

## Patient and public involvement statement

Participants were not directly involved in planning the study; results of this and other HDSS studies are regularly fed back to participants in the HDSS site.

## Statistical analysis

All analyses were done using SPSS V.26 (IBM). We first described all variables using mean and SD, or median and IQR, for normally and non-normally distributed continuous variables, and count and proportion (95% CI) for categorical variables.

We used multivariable logistic regression to explore the associations between the main outcome variables and demographic characteristics or disease states. ORs and 95% CIs are presented. All variables were included in the model. Figures were produced using the R package ggplot2.[47]

Associations between seeking care for an injury or another reason and healthcare experience and health system responsiveness were tested using the Mann-Whitney U test for the non-normally distributed continuous variables. Sample sizes are stated for each analysis.

## RESULTS

The median age of respondents was 52 years (IQR 45–62), females made up 50.7% (1534/3028) of the population, educational attainment was low, with only 15.6% (472/3028) having any schooling at all (table 1).

**Table 1** Demographic information and disease states for those injured or not in preceding 12 months, and for those with or without a disability following injury

| Variable and statistical test for comparisons | | All % sample size=3028, apart from age and marital status (both N=3026), WHODAS (N=3027) and frailty (N=2806) | Not injured in past 12 months (%) Sample size=2796, apart from age, marital status (both N=2794), WHODAS (N=2795) and frailty (N=2607) | Injured in past 12 months (%) Sample size=232, apart from frailty (N=199) | Injured in past 12 months—no disability (%) sample size=127, apart from frailty (N=108) | Injured in past 12 months and suffered a disability (%) sample size=105, apart from frailty (N=91) |
|---|---|---|---|---|---|---|
| Median age (years) (IQR) | | 52 (45–62) | 52 (45–62) | 50 (45–60) | 49 (44–61) | 51 (45–58.5) |
| Female sex (95% CI) | | 50.3 (48.6 to 52.1) | 51.5 (49.7 to 53.4) | 36.2 (30.3 to 42.6) | 40.9 (32.8 to 49.6) | 30.5 (22.5 to 39.8) |
| Marital status (95% CI) | Married/cohabiting | 75.7 (74.1 to 77.2) | 75.4 (73.8 to 77.0) | 78.4 (72.7 to 83.3) | 73.2 (64.9 to 80.2) | 84.8 (76.7 to 90.4) |
| | Other | 24.3 (22.8 to 25.9) | 24.5 (23.0 to 26.2) | 21.6 (16.8 to 27.3) | 25.2 (19.8 to 35.1) | 15.2 (9.6 to 23.3) |
| Socioeconomic status | Quintile 1 | 19.9 | 20.2 | 16.4 | 17.3 | 15.2 |
| | Quintile 2 | 19.8 | 20.1 | 15.9 | 19.7 | 11.4 |
| | Quintile 3 | 20.0 | 19.8 | 22.4 | 19.7 | 25.7 |
| | Quintile 4 | 20.3 | 20.2 | 20.7 | 17.3 | 24.8 |
| | Quintile 5 | 20.0 | 19.6 | 24.6 | 26.0 | 22.9 |
| What is the highest level of education you have completed? (95% CI) | No formal schooling | 84.4 (83.1 to 85.7) | 84.9 (83.5 to 86.2) | 78.4 (72.7 to 83.3) | 72.4 (64.1 to 79.5) | 85.7 (77.8 to 91.2) |
| | Any schooling | 15.6 (14.3 to 16.9) | 15.1 (13.8 to 16.5) | 21.6 (16.8 to 27.3) | 27.6 (20.5 to 35.9) | 14.3 (8.9 to 22.2) |
| Patient Health Questionnaire depression score (95% CI) | Normal or mild | 91.9 (90.9 to 92.9) | 92.0 (91.0 to 93.0) | 91.0 (86.6 to 94.0) | 90.6 (84.2 to 94.5) | 90.5 (84.5 to 95.4) |
| | Moderate to severe* | 8.1 (7.1 to 9.1) | 8.0 (7.0 to 9.0) | 9.0 (6.0 to 13.4) | 9.4 (5.5 to 15.8) | 8.5 (4.6 to 15.5) |
| Normalised WHO Disability Assessment Schedule 2.0 (median score IQR) | | 8.3 (2.1 to 22.9) | 8.3 (2.1 to 22.9) | 8.3 (2.1 to 25) | 8.3 (2.1 to 22.9) | 10.4 (2.1 to 27.1) |
| Normalised WHO Quality of Life (median and IQR) | | 59.4 (46.9 to 65.6) | 59.4 (46.9 to 59.4) | 56.3 (43.8 to 56.3) | 56.3 (46.9 to 62.5) | 56.3 (40.6 to 65.6) |
| Generalised Anxiety Disorder score (95% CI) | Abnormal | 11.6 (10.5 to 12.8) | 10.6 (9.5 to 11.8) | 23.3 (18.3 to 29.1) | 22.0 (15.7 to 30.0) | 24.8 (17.5 to 33.8) |
| | Normal | 88.4 (87.3 to 89.5) | 89.4 (88.2 to 90.5) | 76.7 (70.9 to 81.7) | 78.0 (70.7 to 84.3) | 75.2 (66.2 to 82.5) |
| Fried Frailty Score (95% CI) | Robust | 45.3 (43.4 to 47.1) | 45.3 (43.4 to 47.2) | 44.7 (38.0 to 51.7) | 38.0 (29.4 to 47.4) | 52.7 (42.6 to 62.7) |
| | Prefrail/Frail | 54.7 (52.9 to 56.6) | 54.7 (52.8 to 56.6) | 55.3 (48.3 to 62.0) | 62.0 (52.6 to 70.6) | 47.3 (37.3 to 57.4) |

*Score of 10 or more.
WHODAS, WHO Disability Assessment Schedule.

**Table 2** Logistic regression model for dependent variable of injury in last 12 months

| | | OR | 95% CI | P value |
|---|---|---|---|---|
| Age | | 0.973 | 0.956 to 0.991 | 0.003 |
| Sex | Male (ref) | | | |
| | Female | 0.436 | 0.310 to 0.613 | <0.001 |
| Marital status | Married or cohabiting (ref) | | | |
| | Not married or cohabiting | 1.247 | 0.812 to 1.917 | 0.313 |
| Wealth quintile | Quintile 1 (ref) | | | |
| | Quintile 2 | 1.011 | 0.597 to 1.712 | 0.967 |
| | Quintile 3 | 1.453 | 0.886 to 2.382 | 0.138 |
| | Quintile 4 | 1.367 | 0.823 to 2.272 | 0.227 |
| | Quintile 5 | 1.795 | 1.079 to 2.985 | 0.024 |
| Educational level | No formal schooling (ref) | | | |
| | Any schooling | 1.107 | 0.749 to 1.636 | 0.609 |
| Patient Health Questionnaire depression score | Normal or mild (ref) | | | |
| | Moderate or severe | 0.559 | 0.295 to 1.059 | 0.075 |
| Generalised Anxiety Disorder score | Normal (ref) | | | |
| | Abnormal | 2.921 | 1.963 to 4.347 | <0.001 |
| Normalised WHO Quality of Life score | | 0.981 | 0.97 to 0.993 | 0.002 |
| Normalised WHO Disability Assessment Schedule 2.0 (0–100) | | 1.004 | 0.992 to 1.016 | 0.512 |
| Frailty | Robust (ref) | | | |
| | Pre-frail/frail | 0.967 | 0.704 to 1.327 | 0.834 |

N=2803.

Of those completing the survey, 7.7% (232/3028) reported suffering an injury in the preceding 12 months (table 1). Of 232 injured in the past 12 months, 105 (45.3%) suffered a disability. In multivariable analyses, younger age, male sex, highest wealth quintile, an abnormal GAD score and lower WHOQOL score were all associated with injury (table 2).

The most common mechanism of injury was being struck or hit by an object, 32.8% (76/232) (figure 1). Of those who suffered a fall, 34.6% (9/26) fell from higher than ground level. Exploratory analysis of the association between the mechanism of injury and wealth (online supplemental appendix tables 1 and 2) suggested that the greater odds of being injured in the higher wealth quintile is related to a greater number of falls from a motorcycle or bicycle in this group; 35.8% (19/53) of those falling from a motorcycle or bicycle were from wealth quintile 5 compared with 5.7% (3/53) in quintile 1 (OR 5.83, 95% CI 1.58 to 21.43).

Falling from a motorcycle or bicycle was the mechanism which most frequently resulted in a disability 27.6% (29/105) (figure 1). In multivariable analysis (table 3), only education was significantly negatively associated with odds of disability (OR 0.407, 95% CI 0.17 to 0.997). Compared with being struck or hit by an object, disability was more common among those falling (OR 6.4, 95% CI 1.896 to 21.602, p=0.003), falling from a motorbike (OR 3.335, 95% CI 1.429 to 7.78, p=0.005) and other (OR 10.755, 95% CI 3.471 to 33.323, p<0.001).

Across all survey participants, 3.9% (119/3028) of people reported their last reason for seeking care was for an injury. These 119 respondents reported high levels of satisfaction with care following injury (figure 2). Ninety-six per cent reported being somewhat or very confident of receiving effective treatment if sick tomorrow, 95% reported a good or better opinion of care provider's skills and knowledge and 90% reported that their needs were met well or better. There were no significant differences in these measures between people seeking care for injuries and those seeking care for other reasons (figure 2 and online supplemental appendix table 3).

Those seeking care following injury reported shorter median wait times (10 min vs 20 min, p=0.002) before consultation and longer consultation times (20 min vs 15 min p=0.002) than those seeking care for another reason. Those seeking care for injury were also more likely to be asked to return to the health facility at a later date, 81.4% (95% CI 73.1% to 87.9%) vs 54.8% (95% CI 49.9% to 53.6%) (figure 2). There was a non-significant trend for those seeking injury care to be more likely to borrow or sell to pay for the care episode, compared with those seeking care for other reasons, 21.2% (95% CI 14.7% to 29.7%) vs 14.3% (95% CI 13.0% to 15.6%).

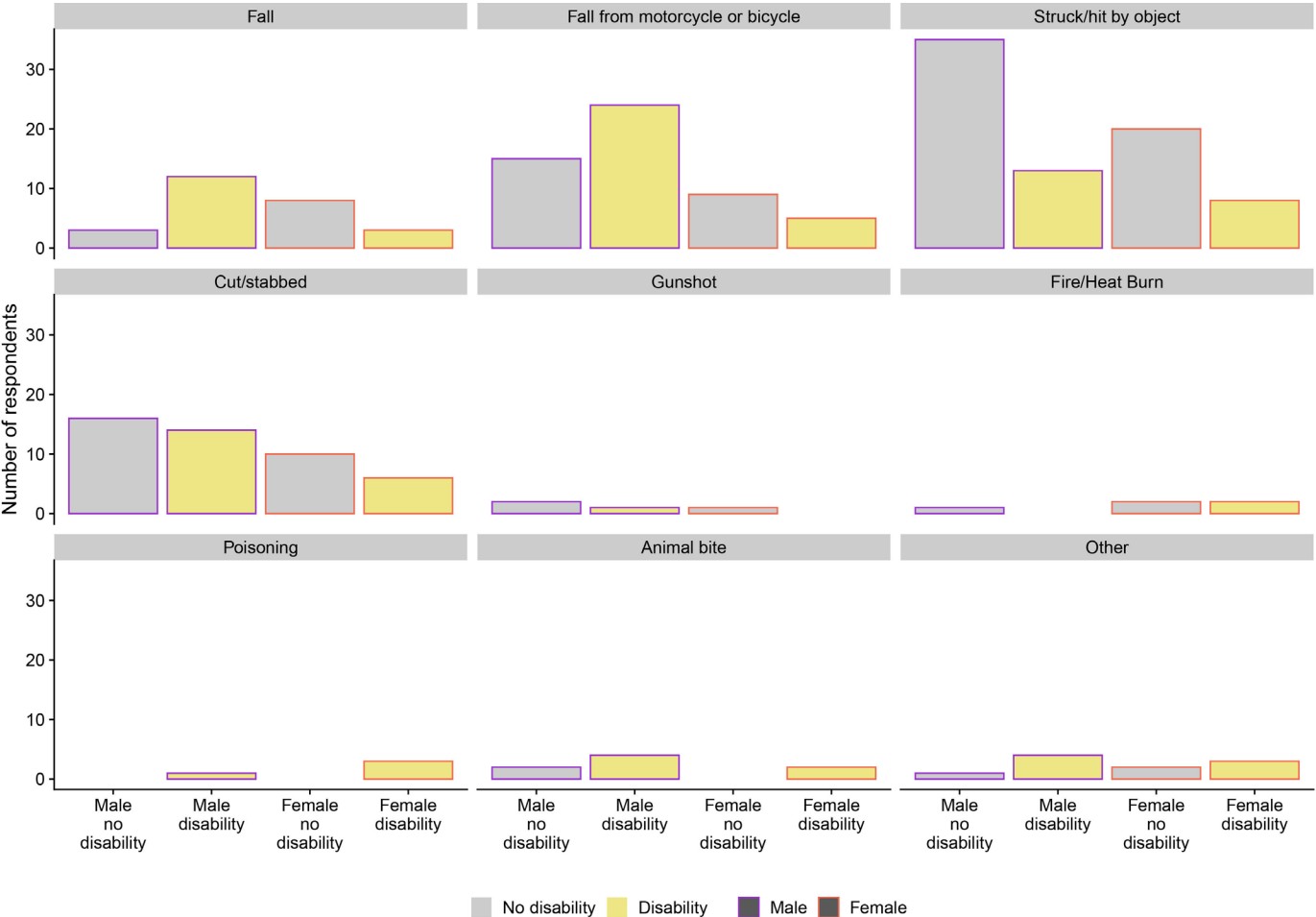

**Figure 1** Mechanism of injury according to sex and associated disability.

## DISCUSSION

This study demonstrates that injuries are prevalent in this older adult rural LMIC population. In this cross-sectional study, injuries were more prevalent in those who were male, or of younger age, or wealthier socioeconomic status; the latter is possibly linked to motorcycle ownership. Those with injuries were more likely to suffer from anxiety, or report a worse quality of life. Almost half of those reporting an injury reported disability as a consequence, which was more common in males and those with lower educational attainment, but not associated with frailty. However, in this cross-sectional survey, we are unable to demonstrate causality. Patient satisfaction with the health system for treatment of an injury was generally high. Having to sell or borrow to pay for care was more common than for non-traumatic healthcare visits, although this did not reach significance. There is little empirical data published on injury prevalence and care within Burkina Faso, particularly in older people. This study can aid researchers and policymakers in understanding the burden to address prevention and avenues for further research.

Globally, poorer populations bear increased injury burden,[48] including among urban populations[49] and those sustaining unintentional injury.[15] This findings is

perhaps due to those of lower SES being exposed to less safe working conditions. Interestingly, we found SES to be positively associated with injury occurrence; potentially, in this rural context, it is likely that relative wealth provided access to motorcycles or bicycles that may have been unaffordable for poorer groups. Further research to prove this hypothesis could have implications for road safety initiatives, particularly if access to motorised transport increases.

Although in an older population, this study found the incidence of injury, 7.7%, was comparable to other sub-Saharan African settings such as in rural Tanzania, rural Rwanda, rural Nigeria, Sudan, Sierra Leone and Kenya where studies have shown prevalence ranges from 4.3% to 15.2%.[15 49–53] Other studies from sub-Saharan Africa have also found injuries to be more common in younger[15 53] or male[15 49 50 52] members of the population. Indeed, male sex is consistently associated with injury globally, with multiple possible contributing factors including alcohol use, dangerous occupations or risk-taking behaviour[54]—unfortunately, none of these were evaluated in CHAS.

Anxiety and reduced quality of life were associated with the occurrence of injury although no association was seen with depression. While this cross-sectional survey could not demonstrate causality, others have shown adverse

**Table 3** Multivariable logistic regression model for dependent variable of disability following injury in last 12 months including mechanism of injury

| | | OR | 95% CI | P value |
|---|---|---|---|---|
| Age | | 1.016 | 0.977 to 1.057 | 0.426 |
| Sex | Male (ref) | | | |
| | Female | 0.471 | 0.215 to 1.033 | 0.06 |
| Marital status | Married or cohabiting (ref) | | | |
| | Not married or cohabiting | 0.433 | 0.15 to 1.25 | 0.122 |
| Wealth quintile | Quintile 1 (ref) | | | |
| | Quintile 2 | 0.636 | 0.187 to 2.167 | 0.469 |
| | Quintile 3 | 1.09 | 0.368 to 3.226 | 0.876 |
| | Quintile 4 | 2.05 | 0.666 to 6.306 | 0.211 |
| | Quintile 5 | 1.521 | 0.503 to 4.596 | 0.457 |
| Educational level | No formal schooling (ref) | | | |
| | Any schooling | 0.407 | 0.166 to 0.997 | 0.049 |
| Patient Health Questionnaire depression score | Normal or mild (ref) | | | |
| | Moderate or severe | 0.426 | 0.107 to 1.689 | 0.225 |
| Generalised Anxiety Disorder 2.0 score (0–100) | Normal (ref) | | | |
| | Abnormal | 0.823 | 0.36 to 1.882 | 0.644 |
| Normalised WHO Quality of Life score | | 1.012 | 0.985 to 1.039 | 0.387 |
| Normalised WHO DAS 2.0 score (0–100) | | 1.023 | 0.997 to 1.05 | 0.084 |
| Frailty | Robust (ref) | | | |
| | Prefrail/frail | 0.562 | 0.284 to 1.112 | 0.098 |
| Mechanism of injury | Struck or hit by object (ref) | | | |
| | Fall | 6.4 | 1.896 to 21.602 | 0.003 |
| | Fall from motorbike | 3.335 | 1.429 to 7.78 | 0.005 |
| | Cut or stabbed | 2.426 | 0.949 to 6.201 | 0.064 |
| | Other | 10.755 | 3.471 to 33.323 | <0.001 |

N=199.
WHODAS, WHO Disability Assessment Schedule.

mental health outcomes to be sequelae of physical injury and include post-traumatic stress disorder, depression and anxiety.[16 17 55] While research exists in high-income settings, further research into the adverse mental health associations with injury in this and other LMIC contexts is warranted. Studies are needed to both establish the scale of burden, whether associations are causal and the direction of the relationship. Development of culturally specific tools for evaluating post physical trauma mental health in African populations is also required.[56]

Almost half those injured reported disability (although not defined) as a consequence of their injury, and reporting disability after injury was associated with lower educational status.

Other sub-Saharan African studies have reported varying levels of disability after injury, for example, 31.7% in Rwanda,[51] and 11% in Sudan.[57] Disability can be more prevalent in rural compared with urban settings,[58] among the uneducated,[57] and in adults over 60.[58] The different

questions employed in these studies makes direct comparison difficult although the high incidence of disability we found may be due to studying an older population with less physical reserve.

In our study, no association between disability and frailty was seen in the population who had been injured. A lower baseline physical function may affect the threshold for self-reported disability. The non-frail population, with low educational attainment, may also have been more dependent on physical labour than other studies and thus more sensitive to limitations to physical function. The association between disability and lower levels of education seen in our study supports this.

The most common injury mechanisms were being struck or hit by an object, falling from a motorcycle or bicycle, being cut or stabbed or falling. In rural environments, injuries can commonly be a consequence of agricultural activity. This was the leading contributor to injuries in rural Ghana.[58] In Tanzania, cuts or stabs were

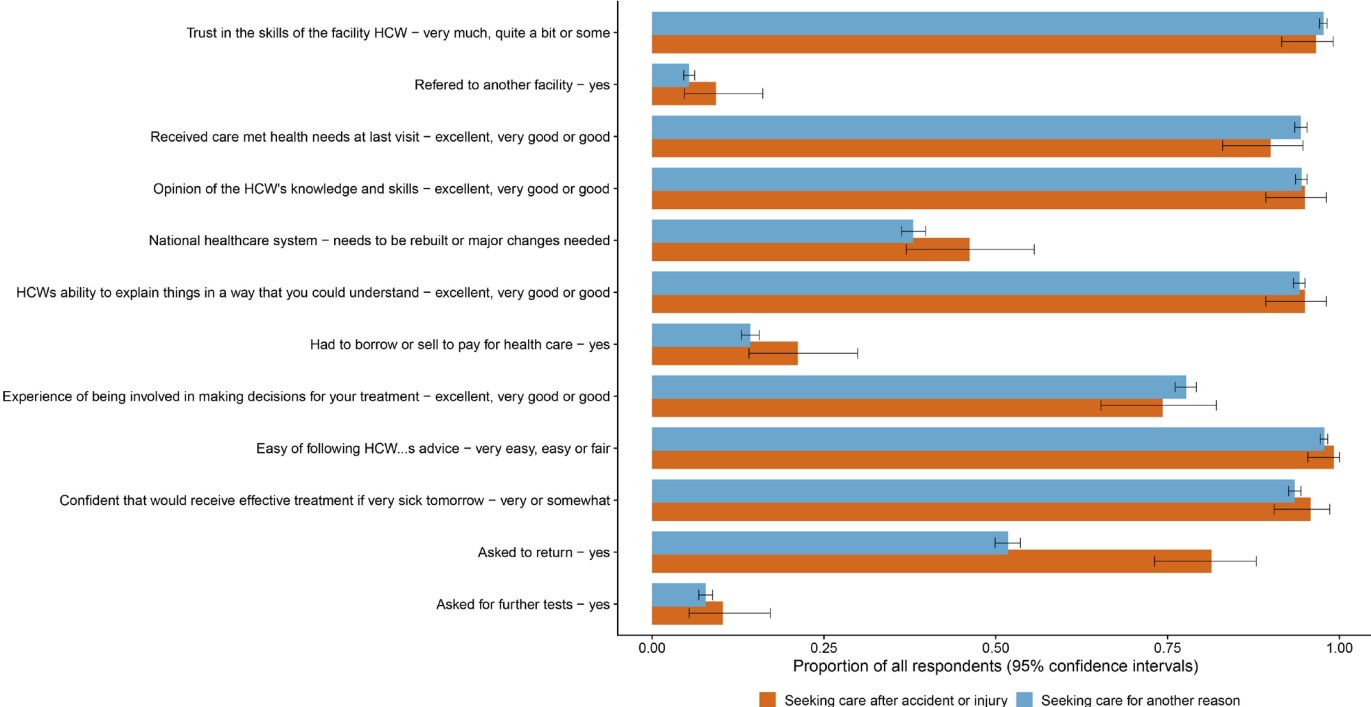

**Figure 2** Opinion and experience of healthcare received by those seeking care following an injury and those seeking care for another reason at last visit. HCW, healthcare worker.

the most common mechanisms in the rural population studied, and two-thirds of cuts were due to agricultural activity.[50] Transport or RTCs were not a distinct category within our study, given that cars are rarely used in Nouna.

Interestingly, despite the older age of the CHAS population relative to most previous studies, falls were relatively uncommon, unlike in Kenya,[15] Nigeria[52] and Sierra Leone[53] and especially in older persons in Tanzania (aged over 60)[50] Ghana (aged over 60)[58] and Sudan (aged over 45).[49] The prevalence of frailty in this population is similar to that seen in other sub-Saharan African populations where this has been studied, so the relatively low prevalence of falls (which are associated with frailty) need further investigation.[32]

Patients reported a positive experience and high satisfaction with care for both injury and non-injury-related consultations. Injured patients experienced shorter wait times and longer consultation time with more frequent requests to return to care than those presenting to healthcare facilities for other reasons. This possibly reflects the urgency of injury care and the need for procedural management such as suturing needing follow-up. Patient satisfaction is influenced by factors such as accessibility, cost, expectation, immediate outcomes and gratitude.[28] However, in LMICs, patient-reported satisfaction may not correspond well with other measures of care quality like safe clinical practice or clinical outcomes. For example, high rates of care satisfaction have been reported, across multiple LMICs, with consultations in which most essential clinical actions were not performed.[28]

As a time critical condition in an economically poor population there is a risk of injury causing impoverishment. Out-of-pocket expenditure for healthcare is commonplace in Burkina Faso.[59] The economic burden of trauma and injuries in LMICs is high through direct medical costs such as medicines, non-medical costs such as transport and indirect costs such as loss of income.[60] Significant economic benefit could derive from reducing injury burden.[61] In Burkina Faso, CT scan access in Ouagadougou was limited by lack of funding when indicated in 20% of cases.[10] Indeed, perceived and actual costs of care are a well noted factor in delaying access to quality care after injury.[26 27 51 57 62] In CHAS, we found a non-significant trend that patients more commonly needed to borrow or sell following injury than other conditions. This could be compounded by the effect of injury-related disability on economic productivity. While CHAS did not capture costs directly, others have found spending on injuries in Nouna HDSS to be higher than other conditions such as malaria and chronic disease.[63] For conditions such as injury, which are unpredictable and high cost, community insurance schemes have been mooted as a way of limiting catastrophic expenditure.[64] Attempts to introduce such schemes in Nouna have suffered high dropout rates, possibly driven by fears around high cost and poor quality of health services.[65] There is evidence from Nouna that individuals enrolled on health insurance schemes received poorer quality healthcare services.[66] Such schemes, limited by low enrolment and selection bias, have failed to make a difference to overall population health including mortality.[67 68]

## Limitations

This study has several limitations. The study was cross-sectional, which limits the causal interpretation of our findings. Injury severity was not assessed, nor was a clear definition of injury or disability provided. No minimum severity or clear definition can lead to inclusion of trivial injuries, overestimation of burden and a lack of comparability across studies.

The time of injury was not reported. Recall loss is well established for community injury household surveys, particularly with respect of less severe injuries (causing less than 30 days disability).[50 69 70] Some studies, therefore, use shorter recall time frames for their survey,[52] or extrapolate minor injuries from the last 30 days reported incidence to calculate annual incidence for minor injuries.[58] Observing, confirming or correcting for recall was not possible in this study and we, therefore, may have underestimated the true incidence. As some injuries are known to be seasonal in Burkina Faso, this known recall bias may mean such injuries could have been misrepresented.[8]

Intentional injuries and domestic violence were not specifically differentiated from being struck or cut. Similarly, RTC was not a separate category, limited to falling from a motorcycle or bicycle, perhaps justified by the low development status of rural Nouna. As RTCs are the main cause for Urban referral facility trauma care in Burkina Faso[9 10] and the only injury related SDG (SDG 3.6)[71] distinguishing these categories could allow future studies to compare within and across countries to inform preventative lessons and strategies.

This study was limited to a rural population. Urban and rural populations in sub-Saharan Africa experience differing burdens of injury.[49 50] Future urban comparisons could add perspective to inform national preventative and research strategies. This study was also confined to older adults and the injury burned for children and young adults is unknown. Fatal injuries were also not captured.

To build on these findings future research focused on injuries could include fatal injuries, across the full population, with rural and urban comparison, capturing the time of injury relative to interview, with specific definitions for injury, disability and mechanism of injury categories, and matched to health system utilisation. This would help further understand the burden in Burkina Faso to inform preventative lessons and strategies as well as plan health system response. Nevertheless, this study adds valuable insight into a relatively under researched topic in a country where little about injury burden or healthcare experience is known.

## Conclusion

This study has demonstrated the importance of injury burden in this older adult rural LMIC population contributing to the limited available published literature on this subject. Further research could inform preventative strategies including safer farming methods and the role of RTCs, enable better understanding the association between adverse mental health and injury in this population, and strengthen health system readiness to provide quality care.

**Author affiliations**
[1]Institute of Applied Health Research, University of Birmingham, Birmingham, UK
[2]King's Centre for Global Health and Health Partnerships, School of Population Health & Environmental Sciences, Faculty of Life Sciences and Medicine, King's College London, London, UK
[3]Academic Department of Military Surgery and Trauma, Royal Centre for Defence Medicine, Birmingham, UK
[4]Institute for Global Health, University College London, London, UK
[5]Department of Epidemiology, Harvard University T H Chan School of Public Health, Boston, Massachusetts, USA
[6]Africa Health Research Institute, KwaZulu-Natal, South Africa
[7]MRC/Wits Rural Public Health & Health Transitions Research Unit (Agincourt), University of the Witwatersrand School of Public Health, Johannesburg, South Africa
[8]Centre de Recherche en Sante de Nouna, Nouna, Burkina Faso
[9]Department of Medical Social Sciences, Northwestern University Feinberg School of Medicine, Chicago, Illinois, USA
[10]Division of Infectious Diseases, Massachusetts General Hospital, Harvard Medical School, Boston, Massachusetts, USA
[11]Heidelberg Institute of Global Health (HIGH), Faculty of Medicine and University Hospitals, University of Heidelberg, Heidelberg, Germany
[12]Centre for Global Surgery, Department of Global Health, Stellenbosch University, Stellenbosch, Western Cape, South Africa

**Contributors** TB, AS and GH conceived and designed the overall CSRN CHAS study. MB and GH coordinated baseline data collection and preparation. JD, JM-G and LRH contributed to the design of the CSRN CHAS household survey. JW and JD designed the current study. JW conducted the analysis, wrote and revised the manuscript. JD supervised the analysis, write up and development of the manuscript. All authors substantively reviewed manuscripts, inputted into revisions and approved the final manuscript.

**Funding** Funding Support for the CRSN Heidelberg Aging Study and for TB was provided by the Alexander von Humboldt Foundation through the Alexander von Humboldt Professor award (no grant number exists) to TB, funded by the German Federal Ministry of Education and Research. GH is supported by a fellowship (210479/Z/18/Z) from both the Wellcome Trust and Royal Society. JM-G was supported by Grant Number T32 AI007433 from the National Institute of Allergy and Infectious Diseases. JW is a serving member of the UK Defence Medical Services.

**Competing interests** None declared.

**Patient consent for publication** Not required.

**Ethics approval** Ethical approval was obtained from Ethics Commission I of the medical faculty Heidelberg (S-120/2018), the Burkina Faso Comité d'Ethique pour la Recherche en Santé (CERS) in Ouagadougou (2018-4-045) and the Institutional Ethics Committee (CIE) of the CRSN (2018-04). Oral assent was sought from all village elders.

**Provenance and peer review** Not commissioned; externally peer reviewed.

**Data availability statement** Data are available on reasonable request. Data may be obtained from a third party and are not publicly available. Data are not publicly available as consent was not given by participants for this to take place. This is in part because entire age cohorts of some villages are included in the data set, potentially allowing for deductive disclosure with sufficient local information. For this reason, anonymised data are available from CHAS study data controllers only following signature of a data use agreement restricting onward transmission. Anyone wishing to replicate the analyses presented or conduct further collaborative analyses using CHAS (which are welcomed and considered based on a letter of intent), should contact GH (g.harling@ucl.ac.uk) in the first instance.

terminology, drug names and drug dosages), and is not responsible for any error and/or omissions arising from translation and adaptation or otherwise.

**ORCID iDs**
John Whitaker http://orcid.org/0000-0001-5877-4496
Guy Harling http://orcid.org/0000-0001-6604-491X
Lisa R Hirschhorn http://orcid.org/0000-0002-4355-7437

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
