## [Reviewer comments · BMJ Open]

ARTICLE DETAILS

TITLE (PROVISIONAL)	Non-fatal injuries in rural Burkina Faso amongst older adults, disease burden and health system responsiveness, a cross-sectional household survey.
AUTHORS	Whitaker, John; Harling, Guy; Sie, Ali; Bountogo, Mamadou; Hirschhorn, Lisa; Manne-Goehler, Jennifer; Bärnighausen, Till; Davies, Justine

VERSION 1 – REVIEW

REVIEWER	Molly Jarman, PhD Brigham and Women's Hospital, Boston, MA, USA
REVIEW RETURNED	12-Nov-2020

GENERAL COMMENTS	Thank you for the opportunity to review the manuscripts describing the epidemiology of injury in rural Burkina Faso. This is important early work to establish need for injury prevention and trauma systems development. The manuscript is well written and the methodological approach is appropriate for the stated research objectives. I do not have any revisions to request/suggest.
---

REVIEWER	Sonya Davey Harvard Medical School / Brigham and Women's Hospital
REVIEW RETURNED	03-Dec-2020

GENERAL COMMENTS	BMJ Open Abstract: "Across all survey participants 3.9% reported seeking care last following injury" Please calculate this percentage from those who were injured. "Among participants who were injured, XX% reported seeking care after their injury" "Injured patients were also asked to return to health facilities more often, 81.4% (95%CI 73.1%–87.9%) vs. 54.8% (95%CI 49.9%–53.6%)" More often than what other group? Please specify the comparator group here Introduction: "This study aimed to evaluate the epidemiology of injury as well as patient-reported health system responsiveness following injury and how this compares with non-injured patient experience, in a population of older individuals in rural Burkina Faso."
--

	The introduction touches on a variety of topics. Please include more information about the goal of this study as it relates to the intro in more detail. Methods: The study design is confusing. Were the authors of this study conducting interviews or is the data entirely from the CRSN Heidelberg Aging Study? Why is the health system experience data not necessarily the same as those injured in the previous 12 months? Is all of the data self-reported from the patients? It would be helpful to have a figure to help illustrate the different data sets used for this study. Results: “In multivariable analyses, younger age, male sex, highest wealth quintile, an abnormal GAD score and lower WHO QOL score were all associated with injury.” Is the abnormal GAD and lower WHO QOL measured before or after the person’s injury? This would be important because I’m trying to understand if it is a demographic component or that the odds of lower GAD and QOL were worse for those who were injured than those who were not injured. “In multivariable analysis (Table 3) only education was significantly negatively associated with odds of disability (Odds ratio 0.407, 95% CI 0.17 – 1.00).” With a 95% OR hitting 1.0, this is not significant. Would remove this statement. Discussion: Sentence 2: please remove - “we cannot show causation, however” “Those with injuries were more likely to suffer from anxiety or depression, or report a worse quality of life.” This statement is not clearly demonstrated within the data. Also, please expand on this position of the discussion. Paragraph 1: Please add the insight about higher SES and motorcycle/bicycle ownership. Would be a great point to make in the discussion “However, in LMICs, patient reported satisfaction may not correspond well with other measures of care input, process, or clinical outcome and has even been associated with poor technical quality care” Please expand more on this - it is a confusing statement. Please consider removing Paragraph 10 of the discussion. Because your study does not assess care quality. Instead, you could highlight that disability post injury is higher in LIC than HIC. Limitations: The limitations section is excellent. Please consider moving some of the details within the limitations section into the methods. For example: -CHAS injury data was self-reported and injuries were not independently verified
--	--

	-There were two injury cohorts described in this study, one reporting annual injury incidence and characteristics, the other reporting last health care visit for treatment following injury.
--	---

REVIEWER	Dr Santosh Bhatta University of the west of England, UK
REVIEW RETURNED	25-Jan-2021

GENERAL COMMENTS	The authors have collected a wealth of data in this study. The study introduction, results and discussion highlight the burden of injury in older adults in a rural area of developing countries—however, there are some concerns with the manuscript in its current form. Title - I would suggest adding the population (population of older individuals/older population) in the title to make it more specific about the study population. Method Definition/ examples of injury and disability, probably under the section Definition of disease states, would help the reader. Though this has been mentioned as a limitation, I would suggest adding this information. Results Patient and public involvement statement – Apart from patient, were there any relevant groups and key personnel involved in designing this study? If so, please mentioned that and need to be thanked in the acknowledgements. Suggest adding - Of 232 injured in the past 12 months, 105 (45.3%) suffered a disability in page 8, line 9. Here, I couldn't understand how you define Younger adult, older adults? Was there any age categories for this? Please justify this. Discussion Page 9, line 25 – It would be better to use the word “unintentional injury” instead of “accidental injury”. Accident/injury – accident/injury has been used in many places. Could you please justify why you used these two worlds together and why not only “injury”? For example, in figure 2 “seeking care after accident or injury,” this could be written, “seeking care after sustaining injury”. Please see: https://www.ncbi.nlm.nih.gov/pmc/articles/PMC1120417/ Page 10, line 16 and 17 or Page 9, line 14 and 15 – Suggest adding specific implications of the findings to practitioners and policymakers if relevant. Some abbreviation and typo – page 5, line 13: Using the first time abbreviation should be defined in full such as DALYs. Page 2, line 50 “he”. Should be “the” etc. Page 13 Table 1,2 and 3 (wherever applicable) – Suggest adding a footnote to define PHQ = Patient Health Questionnaire and GAD = Generalised Anxiety Disorder Pages 23 and 24 – Figure 1 and Figure 2 are quite hard to understand. This figures could be improved with adding legends and choosing contrasting colours.
---

VERSION 1 – AUTHOR RESPONSE

Reviewer: 1

Dr. Molly Jarman, Brigham and Women's Hospital Biomedical Research Institute

Comments to the Author:

Thank you for the opportunity to review the manuscripts describing the epidemiology of injury in rural Burkina Faso. This is important early work to establish need for injury prevention and trauma systems development. The manuscript is well written and the methodological approach is appropriate for the stated research objectives. I do not have any revisions to request/suggest.

Thank you.

Reviewer: 2

Dr. Sonya Davey, Brigham and Women's Hospital

Comments to the Author:

Abstract:

“Across all survey participants 3.9% reported seeking care last following injury”

Please calculate this percentage from those who were injured. “Among participants who were injured, XX% reported seeking care after their injury”

This is not possible due to the structure of the survey. The question relates to what the last reasons for seeking care was. Accident/injury was one possible survey answer. A denominator of the total number of injured participants is not available. Nor would it be meaningful since a participant could logically have been injured 6 months ago (sought care or not), but also sought care 1 month ago for another reason, which would have been the one described for health system responsiveness questions. We have edited the abstract sentence to include the denominator of all who sought care.

“Across all survey participants 3.9% (119/3028) reported their most recent care seeking episode was following injury, rather than for another condition.”

“Injured patients were also asked to return to health facilities more often, 81.4% (95%CI 73.1%–87.9%) vs. 54.8% (95%CI 49.9%–53.6%)”

More often than what other group? Please specify the comparator group here

This has been changed to:

“Injured patients were also asked to return to health facilities more often than those seeking care for another reason, 81.4% (95%CI 73.1%–87.9%) vs. 54.8% (95%CI 49.9%–53.6%).”

Introduction:

“This study aimed to evaluate the epidemiology of injury as well as patient-reported health system responsiveness following injury and how this compares with non-injured patient experience, in a population of older individuals in rural Burkina Faso.”

The introduction touches on a variety of topics. Please include more information about the goal of this study as it relates to the intro in more detail.

The introduction now concludes as follows:

“This analysis primarily aimed to assess the incidence of non fatal injury and variables associated with this amongst older people in rural Burkina Faso, for which little is currently known. Secondary aims were first to describe the incidence of and variables associated with injury related disability, and second, describe patient reported health system responsiveness following injury.”

Methods:

The study design is confusing. Were the authors of this study conducting interviews or is the data entirely from the CRSN Heidelberg Aging Study?

All the data is from the CRSN survey for which the method is briefly described. We have edited the study design section to clarify.

“This study is a **secondary** analysis of the CRSN Heidelberg Aging Study dataset (CHAS).”

Is all of the data self-reported from the patients?

Mostly, however the Fried frailty score included measures of walking speed and grip strength.

“The Fried frailty score was constructed from questions on weight loss in the past year, self-reported activity and levels of exhaustion, combined with measures of walking speed and grip strength ^{44 45}.”

We have clarified in the study design:

“within each selected household one age-eligible adult was randomly selected to complete the survey, which was administered to them by trained data collectors.”

AND

“Injury data was self-reported and injuries were not independently verified.”

Why is the health system experience data not necessarily the same as those injured in the previous 12 months?

It would be helpful to have a figure to help illustrate the different data sets used for this study.

The questions asked in the survey meant that it wasn't possible to align responses on incidence of injury to access to healthcare . Participants were asked if they had an injury in the past 12 months (primary outcome) and those who reported an injury were also asked whether they suffered a disability as a result (a secondary outcome). Questions on healthcare visits and responsiveness were asked of the entire population, and the question pertained to their last visit (for ease of recall). As such, those who reported an injury in the last 12 months may have sought care more recently for another complaint. This has meant that the denominators for these questions are different. This is discussed as a limitation. The below appendix figure 1 attempts to demonstrate this more clearly, which will be included as an appendix and referenced in the methods.

Results:

“In multivariable analysis (Table 3) only education was significantly negatively associated with odds of disability (Odds ratio 0.407, 95% CI 0.17 – 1.00).”

With a 95% OR hitting 1.0, this is not significant. Would remove this statement.

We have rewritten this to report 3 significant figures, as in table 3, consistent with the odds ratio, of the 95%CI (Odds Ratio 0.407, 95%CI 0.17–0.997) which is significant.

“In multivariable analyses, younger age, male sex, highest wealth quintile, an abnormal GAD score and lower WHO QOL score were all associated with injury.” Is the abnormal GAD and lower WHO QOL measured before or after the person’s injury? This would be important because I’m trying to understand if it is a demographic component or that the odds of lower GAD and QOL were worse for those who were injured than those who were not injured.

This is a cross-sectional survey. The GAD and QOL scores are at the time of this survey, and the question about previous injury were asked at the same time. It is not possible therefore to know whether this psychological morbidity is a consequence of injury, or a cause of it. We have included this as a study limitation.

Discussion:

Sentence 2: please remove - “we cannot show causation, however”

This is affirming the above point that, as a cross sectional study, it is not possible to prove that factors such as lower WHO QOL quality of life scores increase the risk of injury, or vice versa. We feel for honest reporting, this should be mentioned upfront in the discussion, but have modified the sentence to state:

“However in this cross sectional survey, we are unable to demonstrate causality.”

“Those with injuries were more likely to suffer from anxiety or depression, or report a worse quality of life.”

This statement is not clearly demonstrated within the data. Also, please expand on this position of the discussion.

As previously commented on in the results section, abnormal Generalised Anxiety Disorder (GAD) score and lower WHO Quality of Life (QOL) score were all associated with injury. This is the basis of this statement.

We have removed “or depression” and expanded the discussion further

“Anxiety and reduced quality of life were associated with occurrence of injury although no association was seen with depression. Whilst this cross sectional survey could not demonstrate causality, others have shown adverse mental health outcomes to be sequelae of physical injury and include post-traumatic stress disorder, depression and anxiety ^{16 17 55}. Whilst research exists in high income settings, further research into the adverse mental health associations with injury in this and other LMIC contexts is warranted. Studies are needed to both establish the scale of burden, whether associations are causal and the direction of the relationship. Development of culturally specific tools for evaluating post physical trauma mental health in African populations is also required ⁵⁶.”

Paragraph 1: Please add the insight about higher SES and motorcycle/bicycle ownership. Would be a great point to make in the discussion

We have brought this discussion point forward in the discussion and explicitly linked to motorcycle ownership within the first paragraph.

“injuries were more prevalent in those who were male, or of younger age, or wealthier socio-economic status; the latter is possibly linked to motorcycle ownership.”

“Globally, poorer populations bear increased injury burden, ⁵⁴ including amongst urban populations ⁵¹ and those sustaining unintentional injury. ¹⁵ This findings is perhaps due to those of lower SES being exposed to less safe working conditions. Interestingly, we found SES to be positively associated with injury occurrence; potentially, in this rural context, it is likely that relative wealth provided access to motorcycles or bicycles that may have been unaffordable for poorer groups.”

“However, in LMICs, patient reported satisfaction may not correspond well with other measures of care input, process, or clinical outcome and has even been associated with poor technical quality

care”

Please expand more on this - it is a confusing statement.

This now reads:

“However, in LMICs, patient reported satisfaction may not correspond well with other measures of care quality like safe clinical practice or clinical outcomes. For example, high rates of care satisfaction have been reported, across multiple LMICs, with consultations in which most essential clinical actions were not performed ²⁸.”

Please consider removing Paragraph 10 of the discussion. Because your study does not assess care quality. Instead, you could highlight that disability post injury is higher in LIC than HIC.

We have removed paragraph 10 as suggested.

Limitations:

The limitations section is excellent. Please consider moving some of the details within the limitations section into the methods. For example:

-CHAS injury data was self-reported and injuries were not independently verified

This has been moved to the methods section.

-There were two injury cohorts described in this study, one reporting annual injury incidence and characteristics, the other reporting last health care visit for treatment following injury.

We have decided to clarify and refer to this as being two question groups rather than cohorts, as they overlap.

“There were, therefore, two injury question groups in this study. The first to determine annual injury incidence and characteristics, the second to determine those for whom the last health care visit followed an injury. Appendix Figure 1 illustrates how these overlapping but distinct question groups are reported.”

Reviewer: 3

Dr. Santosh Bhatta, University of the West of England

Comments to the Author:

The authors have collected a wealth of data in this study. The study introduction, results and discussion highlight the burden of injury in older adults in a rural area of developing countries—however, there are some concerns with the manuscript in its current form.

Title - I would suggest adding the population (population of older individuals/older population) in the title to make it more specific about the study population.

We have changed the title from: "Non-fatal injuries in rural Burkina Faso, disease burden and health system responsiveness, a household survey."

To:

"Non-fatal injuries in rural Burkina Faso amongst older adults, disease burden and health system responsiveness, a cross-sectional household survey."

Method

Definition/ examples of injury and disability, probably under the section Definition of disease states, would help the reader. Though this has been mentioned as a limitation, I would suggest adding this information.

Both injury and disability were self-reported and not verified. It is not possible therefore to define any further. We do list the mechanisms of injury specifically detailed in the survey.

Results

Patient and public involvement statement – Apart from patient, were there any relevant groups and key personnel involved in designing this study? If so, please mentioned that and need to be thanked in the acknowledgements.

None other than those meeting criteria for, and listed as, authors.

Suggest adding - Of 232 injured in the past 12 months, 105 (45.3%) suffered a disability in page 8, line 9.

This has been added.

Here, I couldn't understand how you define Younger adult, older adults? Was there any age categories for this? Please justify this.

The definition of "old" or "young" is very contextually determined. In the UK, one may be considered an older adult when above the age of retirement. In countries where there is no official retirement age, older may be a cultural construct. For example, in Rwanda it is thought to be when a person can no longer dig their plot (personal communication). We are not aware of any local definitions in Burkina Faso, and chose ≥ 40 years to represent older adults based upon other surveys of aging (for example the HAALSI study in South Africa) and life expectancy in Burkina Faso (which is currently 61 years). Hence adults in this survey are likely to be more than 2/3 of the way through their life course.

Discussion

Page 9, line 25 – It would be better to use the word “unintentional injury” instead of “accidental injury”.

Accident/injury – accident/injury has been used in many places. Could you please justify why you used these two words together and why not only “injury”? For example, in figure 2 “seeking care after accident or injury,” this could be written, “seeking care after sustaining injury”. Please see: <https://eur03.safelinks.protection.outlook.com/?url=https%3A%2F%2Fwww.ncbi.nlm.nih.gov%2Fpmc%2Farticles%2FPMC1120417%2F&data=04%7C01%7Cjohn.k.whitaker%40kcl.ac.uk%7C8cf3bd84d9e9442cf29808d8c1ef71ff%7C8370cf1416f34c16b83c724071654356%7C0%7C0%7C637472581462646238%7CUnknown%7CTWFpbGZsb3d8eyJWljojMC4wLjAwMDAiLCJQIjoiV2luMzliLCJBTil6lk1haWwiLCJXVCi6Mn0%3D%7C1000&sdata=0Gqyy3Ps5q8jlbKWaYZYAP0gHIQIDhTjJnGRmBYtdz8%3D&reserved=0>

We have changed to use the term injury and removed “accident”. “Accident or injury” were the terms used in the survey, however we agree that the report is clearer if accident is removed.

Page 10, line 16 and 17 or Page 9, line 14 and 15 – Suggest adding specific implications of the findings to practitioners and policymakers if relevant.

Page 10 line 16-17 has been deleted in response to another reviewer’s comments.

In relation to the future research and policy we have added the following regarding motorbike use and injury.

“Further research to prove this hypothesis could have implications for road safety initiatives, particularly if access to motorised transport increases.”

Some abbreviation and typo – page 5, line 13: Using the first time abbreviation should be defined in full such as DALYs.

This has been spelt out in full.

Page 2, line 50 “he”. Should be “the” etc.

This has been changed.

Page 13 Table 1,2 and 3 (wherever applicable) – Suggest adding a footnote to define PHQ = Patient Health Questionnaire and GAD = Generalised Anxiety Disorder

We have edited to spell out in full within the table for clarity.

Pages 23 and 24 – Figure 1 and Figure 2 are quite hard to understand. This figures could be improved with adding legends and choosing contrasting colours.

Figure legends are given on a separate page in accordance with instruction to authors but will hopefully match up during the copy editing process. The colour contrasts have been enhanced.

VERSION 2 – REVIEW

REVIEWER	Bhatta, Santosh University of the West of England, Faculty of Health and Applied Sciences
REVIEW RETURNED	13-Feb-2021
GENERAL COMMENTS	Thank you for the opportunity to review this revision. The authors have addressed my previous comments very well.